# Helminth exposure and immune response to the two-dose heterologous Ad26.ZEBOV, MVA-BN-Filo Ebola vaccine regimen

Houreratou Barry[1,2], Edouard Lhomme[2,3,4], Mathieu Surénaud[4,5], Moumini Nouctara[1], Cynthia Robinson[6], Viki Bockstal[6¤], Innocent Valea[1,7], Serge Somda[1,8], Halidou Tinto[1,7], Nicolas Meda[1,9], Brian Greenwood[10], Rodolphe Thiébaut[2,3,4☉] *, Christine Lacabaratz[4,5☉]

1 Centre MURAZ, Institut National de Santé Publique Bobo-Dioulasso, Burkina Faso, 2 Univ. Bordeaux, Inserm, Bordeaux Population Health Research Center, UMR 1219; Inria SISTM team, Bordeaux, France, 3 CHU Bordeaux, Department of Medical Information, Bordeaux, France, 4 Vaccine Research Institute (VRI), Créteil, France, 5 Université Paris-Est Créteil, Faculté de Médecine, INSERM U955, Team 16, Créteil, France, 6 Janssen Vaccines & Prevention B.V., Leiden, Netherlands, 7 Institut de Recherche en Sciences de la Santé/Unité de Recherche Clinique de Nanoro, Burkina Faso, 8 Université Nazi BONI, UFR Sciences Exactes et Appliquées, Bobo-Dioulasso, Burkina Faso, 9 UFR Sciences de la santé, Université joseph Ki Zerbo, Ouagadougou, Burkina Faso, 10 London School of Hygiene & Tropical Medicine (LSHTM), London, United Kingdom

☉ These authors contributed equally to this work.
¤ Current address: ExeVir, Ghent, Belgium
* Rodolphe.thiebaut@u-bordeaux.fr

**Data Availability Statement:** Janssen has an agreement with the Yale Open Data Access (YODA) Project to serve as the independent review panel

## Abstract

### Background

The exposure to parasites may influence the immune response to vaccines in endemic African countries. In this study, we aimed to assess the association between helminth exposure to the most prevalent parasitic infections, schistosomiasis, soil transmitted helminths infection and filariasis, and the Ebola virus glycoprotein (EBOV GP) antibody concentration in response to vaccination with the Ad26.ZEBOV, MVA-BN-Filo vaccine regimen in African and European participants using samples obtained from three international clinical trials.

### Methods/Principal findings

We conducted a study in a subset of participants in the EBL2001, EBL2002 and EBL3001 clinical trials that evaluated the Ad26.ZEBOV, MVA-BN-Filo vaccine regimen against EVD in children, adolescents and adults from the United Kingdom, France, Burkina Faso, Cote d'Ivoire, Kenya, Uganda and Sierra Leone. Immune markers of helminth exposure at baseline were evaluated by ELISA with three commercial kits which detect IgG antibodies against schistosome, filarial and *Strongyloides* antigens. Luminex technology was used to measure inflammatory and activation markers, and Th1/Th2/Th17 cytokines at baseline. The association between binding IgG antibodies specific to EBOV GP (measured on day 21 post-dose 2 and on Day 365 after the first dose respectively), and helminth exposure at baseline was evaluated using a multivariable linear regression model adjusted for age and study group.

for the evaluation of requests for clinical study reports and participant-level data from investigators and physicians for scientific research that will advance medical knowledge and public health. Data will be made available following publication and approval by YODA will be considered of any formal request with a defined analysis plan. The study was conducted under IMI consortium. As Consortium Member, HB was able to directly access the data from EBL2001, EBL3001 and EBL2002 that were necessary to conduct this nested sub-study. For any other/future request, the process to request access through YODA is explained here:https://yoda.yale.edu/request/. The data sharing policy of Janssen Pharmaceutical Companies of Johnson & Johnson is available at https://www.janssen.com/clinical-trials/transparency.

**Funding:** This work was an ancillary study undertaken within the framework of the EBOVAC2 project, which was funded by the Innovative Medicine Initiative (IMI) (http://www.imi.europa.eu/). This is a European Union programme (Grant number 115861). The present work is part of a PhD programme within the EBOVAC2 project (Grant number 115861). The PhD budget was already allocated to HB's home institution (Centre MURAZ), which is a member of the consortium put in place to implement the project and conduct a phase 2 trial of the Ad26ZEBOV/MVA-BN-Filo vaccine in Europe and Africa. Both HB and RT, the student and the director of the thesis have actively participated in the implementation of the Project in Africa and Europe. The present work used some data from the EBOVAC1 (Grant number 115854) project but did not receive direct funding from this grant. The funder (IMI) had no role in study design, data collection and analysis, decision to publish, or preparation of the manuscript.

**Competing interests:** I have read the journal's policy and the authors of this manuscript have the following competing interests: VB and CR were full-time employees of Janssen Vaccines and Prevention at the time of the study and may hold shares of Johnson & Johnson.

Seventy-eight (21.3%) of the 367 participants included in the study had at least one helminth positive ELISA test at baseline, with differences of prevalence between studies and an increased prevalence with age. The most frequently detected antibodies were those to *Schistosoma mansoni* (10.9%), followed by *Acanthocheilonema viteae* (9%) and then *Strongyloides ratti* (7.9%). Among the 41 immunological analytes tested, five were significantly (p < .003) lower in participants with at least one positive helminth ELISA test result: CCL2/MCP1, FGFbasic, IL-7, IL-13 and CCL11/Eotaxin compared to participants with negative helminth ELISA tests. No significant association was found with EBOV-GP specific antibody concentration at 21 days post-dose 2, or at 365 days post-dose 1, adjusted for age group, study, and the presence of any helminth antibodies at baseline.

## Conclusions/Significance

No clear association was found between immune markers of helminth exposure as measured by ELISA and post-vaccination response to the Ebola Ad26.ZEBOV/ MVA-BN-Filo vaccine regimen.

## Trial registration

NCT02416453, NCT02564523, NCT02509494. ClinicalTrials.gov.

### Author summary

Recurrent exposure to parasites may influence the immune response to vaccines, especially in endemic African countries. In this study we aimed to assess the association between immune markers of helminth exposure and the immune response post-vaccination with the Ad26.ZEBOV, MVA-BN-Filo Ebola vaccine regimen in African and European participants who participated in three international clinical trials. Seventy-eight (21.3%) of the 367 participants included in the study, had at least one helminth ELISA positive test at baseline with differences of prevalence between studies and an increased prevalence with increasing age. After adjustment for confounding factors, the study did not show a clear association between immunological markers of helminth exposure and the antibody concentration in response to the Ebola vaccine regimen.

## Introduction

Since its discovery in the Democratic Republic of Congo (DRC) in 1976, the Ebola virus has emerged periodically and infected people in several African countries with a high mortality and morbidity rate [1]. The major outbreak of 2014–2016 in West Africa led to the acceleration of vaccine development against EVD and two Ebola vaccines (the rVSV vaccine developed by Merk and Ad26.ZEBOV, MVA-BN-Filo developed by Janssen) have now been prequalified by the WHO and have been used in recent Ebola outbreaks [2–4].

Before its licensure, the heterologous two-dose Ad26.ZEBOV, MVA-BN-Filo Ebola vaccine regimen was evaluated in several clinical trials including phase 1, 2 and 2b trials in the USA, Africa and Europe [5–12]. Although, the results showed the vaccine regimen to be immunogenic, some geographic variabilities of the immune response were found and these variabilities are not yet explained [13]. In the phase 1 trials conducted in Europe and East Africa, one year after dose 1 immunization, the antibody concentrations of European subjects were statistically

significantly higher than those of East African subjects, with the European mean value being 23% higher than the East African one [13].

The observed lower level in induced immune responses in African countries compared to Europe could be explained by several factors such as human genetic, environmental, or demographic factors. One possible explanation is that recurrent exposure to the parasitic infections that are prevalent in African countries could explain such a decrease [14]. Indeed, studies in both human and animal models have shown that trematode, nematode, and *Plasmodium* infections can lead to decreased efficacy of previous vaccination and an inability to ward off new infection [15–18].

Helminth infections may induce a strong T helper (Th2) 2 response, in addition to inducing regulatory T cells, anti-inflammatory cytokines and other mechanisms that modulate the overall immune response [19–21]. A recent paper shows lower HIV-specific ADCC antibody in people infected with S. mansoni [22]. Many studies have suggested that an activated immune microenvironment prior to vaccination may impede the immune response to vaccines. Muyanja et al [23], found in a study conducted in Entebbe, Uganda and Lausanne, Switzerland that immune activation alters cellular and humoral responses to yellow fever 17D (YF-17D) vaccine in an African cohort. They found that YF-17D-induced CD8+ T cell and B cell responses were substantially lower in immunized individuals from Entebbe compared with immunized individuals from Lausanne. The impaired vaccine response was associated with reduced YF-17D replication and higher frequencies of exhausted and activated NK cells, differentiated T and B cell subsets and proinflammatory monocytes suggesting an activated immune microenvironment in the Entebbe volunteers. Another study that examined the relationship between monocytes and natural killer cells with age, HIV infection and influenza vaccine responses showed a deleterious effect of inflammatory monocytes on antigen-specific vaccine response in HIV infection [24].

In Africa, the major soil-transmitted helminth (STH) infections, hookworm (*Necator americanus* and *Ancylostoma duodenale)*, large roundworms (*Ascaris lumbricoides*), and whipworms (*Trichuris trichiura*) are often co-endemic with schistosome infections, in particular *Schistosoma mansoni* [25,26]. Since 2007, the WHO and its partners have been mobilizing resources and working together to achieve control and elimination of helminths infection. Although some progress has being made, the prevalence of helminth infections remains high in some parts of sub-Saharan Africa. Prevalence varies substantially from one country to another and also within country. STH prevalence in Africa ranges between 5% to 20% and Schistosomiasis varies between 10% to 60% [14,27–33].

There is still limited scientific evidence that exposure to helminths infections can have an impact on either the immediate response to vaccination or the durability of the response, and there are no data yet in the case of Ebola vaccines, though it is mainly intended for populations living in areas endemic for helminth infections. Demonstrating a link between helminths infections and the response to vaccine could impact vaccination programmes and raise awareness of the need for better management of helminth infections in endemic countries. The main objective of this study was to assess the association between the immune markers of helminth exposure at baseline to the Ebola virus glycoprotein (EBOV GP) antibody concentration post-vaccination (21 days post dose two and D365 post first vaccination) in African participants from clinical studies EBL2002 and EBL3001 and European participants from clinical study EBL2001 who received an Ad26.ZEBOV, MVA-BN-Filo vaccine regimen against Ebola virus disease. We hypothesized that the immune response to the helminths could be a marker negatively associated with the response to the Ebola vaccine. The secondary objectives of the study were to evaluate the association between inflammatory/activation factors measured at baseline and the EBOV GP antibody concentration post-vaccination and to evaluate the

association between helminth infections with hyper-eosinophilia at baseline and EBOV GP antibody concentration post-vaccination in the same population as eosinophilia is thought to be a marker of helminth infection [34].

## Methods

### Ethics statement

The Phase 2 UK/France study (EBL2001) protocol and study documents were approved by the French national Ethics Committee (CPP Ile de France III; 3287), the French Medicine Agency (150646A-61), the UK Medicines and Healthcare Products Regulatory Agency (MHRA), and the UK National Research Ethics Service (South Central, Oxford; A 15/SC/0211). The Phase 2 Kenya/Uganda/Burkina Faso/Côte d'Ivoire study (EBL2002) protocol and study documents were approved by local and national independent Ethics Committees and Institutional Review Boards of the participating countries. The Sierra Leonean Phase 2 study (EBL3001) protocol and study documents were approved by the Sierra Leone Ethics and Scientific Review Committee, the Pharmacy Board of Sierra Leone, and the London School of Hygiene & Tropical Medicine ethics committee. All adult participants and parents/legal guardian of adolescents and children supplied written informed consent before enrolment in the main trials. Adolescent and older children also provided written informed ascent before inclusion. Informed consent from the main trials stated the use of participants information and samples for the study and other scientific research related to the vaccine and to Ebola virus disease. The samples of Participants included in the present study were randomly selected from remaining samples of the initial clinical trials separately for each study, study interval group, age group and country. This was based on the calculated sample size (**Text A and Table B** in S1 Appendix).

### Study design

We conducted a sub study nested in clinicals trials. A subset of participants in the EBL2001, EBL2002 and EBL3001 trials in which HIV-negative participants received an active Ebola vaccine regimen consisting of Ad26.ZEBOV as dose 1 followed by MVA-BN-Filo as dose 2 [5–12]. All participants were HIV-negative. EBL2001 and EBL2002 were both phase 2 randomised, observer-blind, placebo-controlled studies in which participants received intramuscular injections of Ad26.ZEBOV, followed 28, 56 or 84 days later by MVA-BN-Filo. In the EBL2001 study (ClinicalTrials.gov Identifier: NCT02416453 and EudraCT 2015-000596-27), 18 to 65 years old healthy adults were included in the United Kingdom and France. The EBL2002 study (ClinicalTrials.gov Identifier: NCT02564523) was conducted in Burkina Faso, Cote d'Ivoire, Kenya and Uganda and enrolled three cohorts; 18 to 70 years old healthy adults, HIV infected adults, 12 to 17 years old adolescents and 6 to 11 years old children. The EBL3001 (ClinicalTrials.gov Identifier: NCT02509494) was a phase 3 double-blinded study which evaluated the safety and immunogenicity of the Ad26.ZEBOV (at day 0) and MVA-BN-Filo (at day 56) vaccine regimen in healthy adults, adolescents, children and toddlers in Sierra Leone.

### Ebola binding antibody assay

The Ebola binding antibody concentration was assessed as a primary or secondary endpoint of the three studies as previously described [5–12]. EBOV GP-specific binding antibodies were measured at all timepoints using an EBOV GP Filovirus Animal Non-Clinical Group (FANG) enzyme-linked immunosorbent assay (ELISA) performed at Q2 Solutions (San Juan Capistrano, CA, USA). For this substudy, IgG binding antibodies specific to EBOV GP at Day 21 post-dose 2, and on Day 365 after the first dose were the two endpoints considered.

## Helminth ELISA testing

A subset of the remaining baseline serum samples (pre vaccination) from the three trials' participants has been used for analysis of helminth IgG and inflammation and activation markers. Helminth IgG was evaluated by ELISA with three commercial kits (ELISA *Schistosoma mansoni*; ELISA *Acanthocheilonema viteae*; ELISA *Strongyloïdes ratti*,) to measure baseline IgG against the pathogens causing schistosomiasis, strongyloidiasis and filariasis respectively (**Table A** in S1 Appendix).

The assays were performed following manufacturer's instructions. Absorbance at 405 nm was read using an ELISA microplate reader. Results were expressed as the ratio of the optical density (OD) of each sample to the OD of the "threshold" sample supplied in each kit. Samples with a ratio > 1 were considered positive. For all the ratios between 0.9 and 1.1, the measurement was repeated to confirm the result. If the result was discordant, a third replicate was performed.

Eosinophilia was defined as a blood eosinophil count higher than 500/ mm3 and hyper eosinophilia as an eosinophil level higher than 1000/mm3. Eosinophil counts were available only for studies EBL2001 and EBL2002.

## Inflammation/activation markers

Luminex technology with commercial kits (Human XL cytokine discovery premixed magnetic Luminex Perf assay kit 41plex, R&D systems) was used to measure inflammatory and activation markers, and Th1/Th2/Th17 cytokines at baseline prior to vaccination. The following 41 markers were measured: CCL2/MCP-1, CCL4/MIP-1β, CCL19/MIP-3β, CD40L/TNFSF5, CXCL10/IP-10, FGF-basic, G-CSF, GRZ B, IFN-β, IL-1α/IL-1F1, IL-1-ra/IL-1F3, IL-3, IL-5, IL-7, IL-10, IL-13, IL-17A, B7-H1/PD-L1, TNF-α, VEGF, CCL3/MIP-1α, CCL11/Eotaxin, CCL20/MIP-3α, CX3CL1/Fractalkine, EGF, Flt-3L, GM-CSF, IFN-α, IFN-γ, IL-1β/IL-1F2, IL-2, IL-4, IL-6, IL-8/CXCL8, IL-12p70, IL-15, IL-33, TGF-α, TRAIL, CCL5/RANTES and IL-17E/IL25. Results were expressed as concentration of each analyte (pg/ml). We included results with extrapolated data (data lower than the last range point, but whose concentration was still calculated by the analysis software). For data below the last range point for which the concentration could not be calculated (<OOR), we imputed the lowest value obtained.

## Statistical aspects

**Sample size.** For the sample size calculation, we assumed a 25% prevalence of helminth exposure at baseline among African participants as the ELISA could detect both past and current infections. For a prevalence of 25%, a total of 28, 24 and 41 participants who had received MVA-BN-Filo at 28, 56 and 84 days post Ad26.ZEBOV, respectively, were needed to demonstrate a minimum difference of antibody titre of 0.56 (on log-scale) between participants positive for helminths exposure compared to those who were antibody negative with a statistical power of 80% and alpha risk of 5%. This minimum difference corresponds to the interquartile range observed at D21 post dose 2 in the EBL2002 study among participants who received dose 2 at D56 post dose 1. The sample size calculations were performed in regards to the primary hypothesis to be tested for each study separately (see more details in Text A in S1 Appendix).

**Statistical analyses.** Spearman correlations were performed between the concentration of inflammatory and activation markers measured at baseline and the EBOV GP antibody concentration 21 days post-dose 2 and 365 days post-dose 1 vaccination, respectively. The association between the immune response to helminths and the levels of activation of inflammatory markers was assessed using a Student's t-test. The association between the immune response

to helminths and eosinophilia, was assessed by the Chi square test. The association between EBOV GP antibodies and eosinophil count was assessed by Pearson correlation and a mean comparison of the concentration of EBOV GP antibodies with normal eosinophilia ($<500/mm^3$) or hyper-eosinophilia (level $>500/mm^3$) was performed using a Student's t-test.

The association between the concentration of EBOV GP antibodies 21 days post-dose 2 or 365 days post-dose 1, and positive ELISA for helminth exposure at baseline (ELISA *S. mansoni*, ELISA *A. viteae*, ELISA *S. ratti*) was evaluated by study (EBL2001, EBL2002 and EBL3001), in the pooled studies for any ELISA test (considered positive if at least one of the three tests was positive) and for each of the three ELISA tests separately using Student's t-test. Linear regression models with the log transformed concentration of EBOV GP antibodies at either 21days post-dose 2 or at 365 day post-dose 1 as the dependant variable were used to adjust for potential confounding factors (age group and study). We analyzed the immune response to helminth exposure as a five modalities variable: no positive test (modality 1), a single positive test (modality 2–4), or multiple positive tests (modality 5). All analyses were conducted using R version 1.2.5033 and a p-value $\leq 0.05$ was considered as statistically significant.

## Results

### Study population

A total of 367 participants was included in the analyses: 87 (23.7%) from EBL2001, 184 (50.1%) from EBL2002 and 96 (26.2%) from EBL3001. Study population characteristics (sex, age, country) are described in **Table 1**.

### Prevalence of helminth exposure at baseline

A total of 78 participants (21.3%) had at least one positive helminth ELISA test at baseline, including 20 (5.45%) participants with at least two positive helminth ELISA tests and 4 (1.1%) participants with all three helminth ELISA tests positive. *S. mansoni* was the most frequent helminth infection detected (40 participants, 10.9%). 33 participants (9.0%) and 29 participants, 7.9%) had positive ELISA tests to *A. viteae* and *S. ratti* respectively. The prevalence of helminth ELISA test positivity is shown in **Fig 1** and Table C in **S1 Appendix**.

**Table 1. Description of the population included in the helminth substudy by trial (EBL2001, EBL2002, EBL3001).**

| Characteristics | Pooled studies (N = 367) | EBL2001 (Europe) (N = 87) | EBL2002 (East and West Africa) (N = 184) | EBL3001 (Sierra Leone) (N = 96) |
|---|---|---|---|---|
| Sex, n (%) | | | | |
| Female | 167 (45.5%) | 46 (52.8%) | 83 (45.1%) | 38 (39.6%) |
| Males | 200 (54.5%) | 41 (47.2%) | 101 (54.9%) | 58 (60.4%) |
| Age, years | | | | |
| median (Q1, Q3) | 19 (10, 3) | 40 (30, 5) | 16 (11, 3) | 10 (4, 2) |
| Adults, n (%) | 192 (52.3%) | 87 (100.0%) | 81 (44.0%) | 24 (25.0%) |
| Adolescents, n (%) | 72 (19.6%) | 0 (0.0%) | 49 (26.6%) | 23 (24.0%) |
| Children, n (%) | 103 (28.1%) | 0 (0.0%) | 54 (29.3%) | 49 (51.0%) |
| Country, n (%) | | | | |
| France | 34 (9.3%) | 34 (39.1%) | - | - |
| United Kingdom | 53 (14.4%) | 53 (60.9%) | - | - |
| Burkina Faso | 55 (15.0%) | - | 55 (29.9%) | - |
| Cote d'Ivoire | 37 (10.1%) | - | 37 (20.1%) | - |
| Kenya | 42 (11.4%) | - | 42 (22.8%) | - |
| Uganda | 50 (13.6%) | - | 50 (27.2%) | - |
| Sierra Leone | 96 (26.2%) | - | - | 96 (100.0%) |

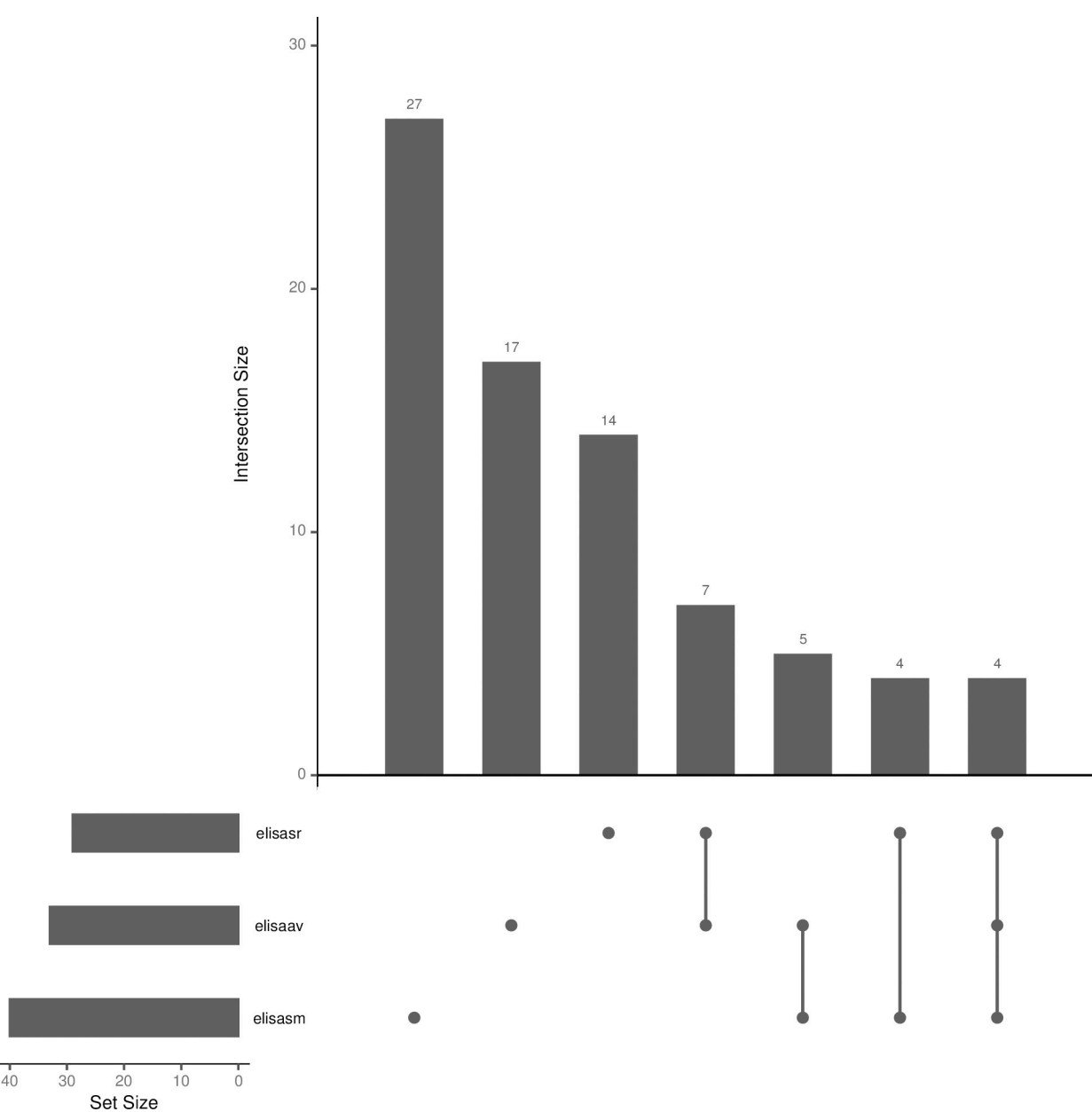

**Fig 1. Description of helminth ELISA tests results at baseline among the 367 participants from EBL2001, EBL2002 and EBL3001 trials.**
Elisasr: This line shows the 29 samples which were positive in the S. ratti ELISA test. Elisaav: This line shows the 33 samples which were positive in the A. viteae ELISA test. Elisasm: This line shows the 40 samples which were positive in the S. mansoni ELISA test. The intersection size on the y axis shows the number of samples which were positive for the tests indicated with dots (single test) or dots and lines (multiple tests) below the x axis. Each bar shows the number of samples positive to each test taken alone and the number of samples positive to all the possible combination of the tests.

The distribution of the type of helminth by age group, study and country is shown in **Table 2.** The positive rate increased with age with 26% (50/192) of adults, 18% (13/72) of adolescents and 15% (15/103) of children having at least one positive helminth ELISA test (p = 0.05). No significant difference was found between males (37/200, 18%) and females (41/167, 25%) (p = 0.20). Adults represented 64% of all participants with at least one positive ELISA test. More participants had at least one positive helminth ELISA test in EBL2002 study

**Table 2. Distribution of the type of helminth by age group (cohort), study and country.**

| Characteristic | None, N = 289 | *A.viteae*[1], N = 17 | *S.mansoni*[1], N = 27 | *S.ratti*[1], N = 14 | Multiple[2], N = 20 | Overall, N = 367 |
|---|---|---|---|---|---|---|
| **AGE GROUP, n (%)** | | | | | | |
| **Children** | 88 (85%) | 4 (3.9%) | 4 (3.9%) | 1 (1.0%) | 6 (5.8%) | 103 (100%) |
| **Adolescents** | 59 (82%) | 3 (4.2%) | 5 (6.9%) | 1 (1.4%) | 4 (5.6%) | 72 (100%) |
| **Adults** | 142 (74%) | 10 (5.2%) | 18 (9.4%) | 12 (6.2%) | 10 (5.2%) | 192 (100%) |
| **SEX** | | | | | | |
| **Female** | 126 (75%) | 11 (6.6%) | 12 (7.2%) | 9 (5.4%) | 9 (5.4%) | 167 (100%) |
| **Male** | 163 (82%) | 6 (3.0%) | 15 (7.5%) | 5 (2.5%) | 11 (5.5%) | 200 (100%) |
| **STUDY, n (%)** | | | | | | |
| **EBL2001** | 78 (90%) | 1 (1.1%) | 1 (1.1%) | 6 (6.9%) | 1 (1.1%) | 87 (100%) |
| **EBL2002** | 137 (74%) | 14 (7.6%) | 18 (9.8%) | 7 (3.8%) | 8 (4.3%) | 184 (100%) |
| **EBL3001** | 74 (77%) | 2 (2.1%) | 8 (8.3%) | 1 (1.0%) | 11 (11%) | 96 (100%) |
| **COUNTRY, n (%)** | | | | | | |
| **Burkina Faso** | 41 (75%) | 2 (3.6%) | 10 (18%) | 0 (0%) | 2 (3.6%) | 55 (100%) |
| **Cote Ivoire** | 25 (68%) | 2 (5.4%) | 4 (11%) | 2 (5.4%) | 4 (11%) | 37 (100%) |
| **France** | 30 (88%) | 1 (2.9%) | 0 (0%) | 2 (5.9%) | 1 (2.9%) | 34 (100%) |
| **United Kingdom** | 48 (91%) | 0 (0%) | 1 (1.9%) | 4 (7.5%) | 0 (0%) | 53 (100%) |
| **Kenya** | 38 (90%) | 0 (0%) | 2 (4.8%) | 2 (4.8%) | 0 (0%) | 42 (100%) |
| **Sierra Leone** | 74 (77%) | 2 (2.1%) | 8 (8.3%) | 1 (1.0%) | 11 (11%) | 96 (100%) |
| **Uganda** | 33 (66%) | 10 (20%) | 2 (4.0%) | 3 (6.0%) | 2 (4.0%) | 50 (100%) |

[1] Positive ELISA test for only this specific helminth test

[2] Multiple positive ELISA tests (a combination of at least two positive helminth ELISA tests)

(47/184, 26%) and in the EBL3001 study (22/96, 23%) as compared to the EBL2001 study (9/87, 10%). The highest positive rate was found in Uganda (34%) followed by Cote d'Ivoire (32%), Burkina Faso (25%) then Sierra Leone (23%). The lowest rates of any positive helminth ELISA test were found in England (9.4%), Kenya (9.5%), and France (12%). Results are shown in **Tables D and E in S1 Appendix.**

## Eosinophil level at baseline

Only 18 participants were found with eosinophilia (an eosinophil count higher than 500 cells / mm3 at baseline, all from the EBL2002 study. Among them, 9 (50%) had at least one positive helminth ELISA test. More participants with at least one positive helminth ELISA test had eosinophilia (9/55, 16.4%) than participants with negative helminth ELISA tests (9/213, 4.2%) (p = 0.003, **Tables F, G, and H** in S1 Appendix). No association was found between EBOV-GP antibody levels and the presence of eosinophilia (**p = 0.56, Fig A in S1 Appendix**).

## Inflammatory markers at baseline

None of the inflammatory and activation markers measured at baseline were correlated with EBOV GP antibody concentration, either at 21 days post dose 2 (**Fig Ba in S1 Appendix),** or at 365 days post dose 1 (**Fig Bb in S1 Appendix**). Among the 41 analytes tested, 5 were significantly lower in participants with at least one positive helminth ELISA test compared to those without any positive test: IL-13 (p = 0.008), CCL11/Eotaxin (p = 0.002), IL-7 (p = 0.004), CCL2/MCP-1 (p = 0.002), and basic FGF (p = 0.002) (**Fig C in S1 Appendix**).

The distributions of the inflammatory/activation markers as measured by Luminex were significantly different between studies, with a general increase of inflammatory and activation

**Table 3. Results of univariable and multivariable linear regression analysis of EBOV-GP antibody concentration (log 10 transformed) at 21 days post dose 2 adjusted for helminth ELISA test, age group and study.**

| Variables | Univariable analysis | | | Multivariable analysis | | |
|---|---|---|---|---|---|---|
| | Coefficient | p-value | 95% CI | Coefficient | p-value | 95% CI |
| **Any helminth (Yes vs No)** | -0.24 | 0.07 | -0.51–0.02 | | | |
| **Helminth** | | 0.18 | | | 0.19 | |
| None | ref | | | | | |
| *A. viteae* | -0.61 | 0.02 | -1.13 –-0.09 | -0.60 | 0.02 | -1.12 –-0.08 |
| *S. mansoni* | -0.15 | 0.47 | -0.57–0.26 | -0.10 | 0.64 | -0.53–0.32 |
| *S. ratti* | -0.02 | 0.95 | -0.5–0.55 | 0.11 | 0.70 | -0.46–0.68 |
| Multiple[1] | -0.22 | 0.37 | -0.69–0.26 | -0.22 | 0.37 | -0.70–0.27 |
| **Age group** | | 0.002 | | | 0.02 | |
| Adults | ref | | | ref | | |
| Children | 0.45 | 0.0004 | 0.20–0.70 | 0.42 | 0.006 | 0.12–0.73 |
| Adolescent | 0.20623 | 0.15 | -0.07–0.49 | 0.17 | 0.29 | -0.15–0.49 |
| **Study** | | 0.23 | | | 0.57 | |
| EBL2001 | ref | | | ref | | |
| EBL2002 | 0.20 | 0.14 | -0.07–0.47 | 0.09 | 0.58 | -0.23–0.40 |
| EBL3001 | 0.25 | 0.11 | -0.05–0.55 | 0.03 | 0.86 | -0.34–0.41 |

[1]Multiple positive ELISA tests (a combination of at least two positive helminth ELISAs tests)

markers (i.e. CCL19/MIP-3β, TRAIL, CCL4/MIP-1β, CXCL10/IP-10, IL-6, IL-1ra, IL-3, IL-5, IL-10, IL-17A, TNFα, IL-33, TGFα, CCL5/RANTES, PD-L1, VEGF, MIP-1α, MIP-3α, CX3CL1) in African trials (EBL2002, EBL3001) as compared to the trial conducted in Europe (EBL2001). All results are provided in **Fig D in S1 Appendix.**

### EBOV-GP antibody concentration at Day 21 post-dose two

In the pooled univariable analysis (**Table 3**), the EBOV-GP antibody concentration 21 days post-dose 2 was not significantly associated with any helminth ELISA tests (any Helminth, **p = 0.07**). although EBOV-GP antibody concentrations tended to be lower in participants with an *A. viteae* positive ELISA test compared to those without any positive ELISA (-0.61 log units/ml, 95% CI: -1.13 –-0.09). The EBOV-GP antibody concentration at 21 days post dose 2 was significantly different according to age groups (p = 0.002). Children had a higher EBOV-GP binding antibody GMC at Day 21 post-dose 2 than adults (+0.45 log unit/ml, 95% CI: 0.20–0.70).

A multivariable analysis adjusted for age group and the study was performed. There was no association between helminth ELISA seropositivity and EBOV-GP antibody concentration at 21 days post-dose 2 (p = 0.19). Age group was still associated with antibody concentration with a higher EBOV-GP antibody concentration at 21 days post dose 2 in children compared to adults (+0.42 log unit/ml, 95% CI: 0.12–0.73).

### EBOV-GP antibody concentration at Day 365 post-dose 1 vaccination

The univariable and multivariable analyses of the EBOV GP antibody concentration at Day 365 found no significant association with helminth exposure (**Table 4**). However, age group and study were independently associated with the EBOV-GP antibody concentration. Adolescence exhibited a better response as compared to children (-0.41 log unit/ml, 95% CI: -0.73

**Table 4. Results of the univariable and multivariable analysis of EBOV-GP antibody concentration (log 10 transformed) at 365 days post-dose 1 vaccination, adjusted for helminth ELISA test, age group and study.**

| Variables | Univariable analysis | | | Multivariable analysis | | |
|---|---|---|---|---|---|---|
| | Coefficient | p-value | 95% CI | Coefficient. | p-value | 95% CI |
| **Any helminth (Yes vs No)** | -0.17 | 0.23 | -0.45–0.11 | | | |
| **Helminth** | | 0.18 | | | 0.85 | |
| None | ref | | | ref | | |
| *A.vitea* | -0.06 | 0.82 | -0.60–0.48 | -0.13 | 0.61 | -0.64–0.38 |
| *S. mansoni* | -0.18 | 0.42 | -0.61–0.26 | -0.10 | 0.64 | -0.52–0.32 |
| *S. ratti* | 0.16 | 0.60 | -0.45–0.78 | -0.09 | 0.77 | -0.67–0.50 |
| Multiple[1] | -0.48 | 0.06 | -0.98–0.02 | -0.24 | 0.32 | -0.72–0.24 |
| **Age** | 0.01 | 0.003 | 0.004–0.01 | | | |
| **Age group** | | 0.005 | | | 0.03 | |
| Adults | ref | | | ref | | |
| Children | -0.22 | 0.11 | -0.49–0.02 | 0.12 | 0.41 | -0.17–0.42 |
| Adolescent | -0.52 | 0.001 | -0.82 – -0.21 | -0.28 | 0.08 | -0.60–0.03 |
| **Study** | | <0.001 | | | <0.001 | |
| EBL2001 | ref | | | | | |
| EBL2002 | -0.50 | 0.003 | -0.82–0.17 | -0.44 | 0.02 | -0.81 – -0.07 |
| EBL3001 | -1.17 | <0.001 | -1.53 – -0.81 | -1.15 | <0.001 | -1.56 – -0.73 |
| **Country** | | <0.001 | | | | |
| **United Kingdom** | ref | | | | | |
| **Burkina Faso** | -0.48 | 0.05 | | | | |
| **Cote d'Ivoire** | -0.74 | 0.005 | | | | |
| **France** | -0.41 | 0.16 | | | | |
| **Kenya** | -0.85 | 0.001 | | | | |
| **Sierra Leone** | -1.37 | <0.001 | | | | |
| **Uganda** | -0.77 | 0.002 | | | | |

[1]Multiple positive ELISA tests (a combination of at least two positive helminth ELISAs)

––0.09). Even after adjustment for helminth exposure and age, the participants included in Europe had a better response than in Africa (p<0.001).

## Discussion

As expected, the prevalence of helminth exposure as indicated by detection of antibodies against *S.mansoni*, *A viteae* and *S ratti* varied according to geographical location, and increased with age. Helminth exposure did not impact the EBOV-GP antibody concentration post Ad26. ZEBOV, MVA -BN-Filo vaccination, neither at 21 days post-dose 2, nor one-year post-dose 1.

The three ELISA tests used in our study allowed us to estimate exposure to the main helminths responsible for schistosomiasis, soil transmitted helminth infections (*Strongyloides*) and filariasis. A positive helminth ELISA test revealed the presence of antibodies against the targeted helminth species that could be the result of a recent or past exposure. Therefore, we may have missed the association if only recent or ongoing exposure to an helminth has an impact on the immune response to the vaccine.

We used the eosinophil count as an indirect measure of recent exposure to helminth infection. However, the prevalence of hyper-eosinophilia was very low, which is consistent with a low prevalence of current or recent infection in helminth-exposed participants in our study. As many pathological conditions can lead to eosinophilia, the common causes being allergies,

drug reactions and parasitosis, it remains an imperfect marker to evaluate a recent helminth infection. The limitation of using these ELISA tests is the existence of high level of cross-reactivity between the ELISA assays and with other helminths. There could be an association with a specific infection that these broad-based assays might not have picked up.

Although no clear association between EBOV-GP antibody concentration post Ad26. ZEBOV, MVA -BN-Filo vaccination and exposure to the three helminths tested was found, other helminth could be associated with the immune response. Indeed, specific helminth-derived products have been characterized as having immunomodulatory effects, including lysophosphatidylserine, phosphatidylserine, lacto-N-fucopentaose III, *Acanthocheilonema viteae* ES62, *Schistosoma mansoni* dsRNA, cathepsin cysteine proteases and others. These products may have effects that function via interactions with Toll-like receptors (TLR), including TLR-2,3, or 4 in particular, or may impact TLR signaling pathways. Certain immunomodulatory effects may also function via TLR-independent mechanisms [21].

In Africa the major soil-transmitted helminth (STH) infections; hookworm (*Necator americanus* and *Ancylostoma duodenale*, large roundworms (*Ascaris lumbricoides*), and whipworms (*Trichuris trichiura*) are often co-endemic with *Schistosoma mansoni*, a trematode parasite [25]. These three helminth tests were chosen in our study for two other reasons in addition to their documented immunomodulatory effect. The first reason was their high prevalence in some of our target countries and the second reason was the availability of the tests. In Sierra Leone, a national school-based survey conducted in 2012 and 2016 to evaluate the progress of Mass Drug Administration against helminth infections found an overall *Schistosoma mansoni* prevalence in the seven MDA districts of 20.4% (95% CI: 18.7–22.3%) [33]. In the "haut Bassin region" of Burkina Faso, this was 8.75% [29].

Other parasites not measured in our study could also play a role in modulating the immune response in Africa such as malaria [35–38]. However, Ishola D et al [38] did not find a consistent effect of malaria infection on EBOV-GP binding antibody concentrations assessed post-dose 2 vaccination with the Ad26.ZEBOV, MVA-BN-Filo regimen in Sierra Leone, but they did observe a trend toward lower vaccine-induced antibody concentrations (GMR, 82; 95% CI, .67–1.02) in participants with malaria infection as measured by microscopy.

A majority of the activation/inflammation biomarkers quantified in this study were increased at baseline in individuals enrolled in African trials (EBL2002 and 3001) as compared to the European trial (EBL2001). This observation is consistent with what was already described in the literature [39–41]. However, we did not observe any correlation between anti-EBOV GP antibody response and any of the biomarkers measured at baseline. Other markers not quantified in this study and more specific of antibody responses, like IL-21, CXCL13, BAFF and APRIL, could be different in African and European individuals. Helminth exposure, as measured in this study, did not seem to be associated with activation/inflammation markers and showed a decrease of IL-13, CCL11/Eotaxin, IL-7, CCL2/MCP-1 and bFGF. Thus, these results are consistent with the absence of an effect of helminth exposure on the level of anti-EBOV GP antibody response observed after vaccination.

In conclusion, the results of the present study do not suggest an association between the immune response to helminths exposure and the response to the Ad26.ZEBOV, MVA-BN--Filo vaccine regimen. Nonetheless, several studies have reported lower vaccine immunogenicity, in developing countries for some vaccines [42–44]. Various hypotheses explaining variability of the immune response have been made, notably genetic factors, environmental factors such as activated immune microenvironment, exposure to environmental pathogens (bacterial, viral, parasitic) microbiome, pre-existing immunity specially for adenovirus and demographic factors such as age, sex and BMI. All these factors are potential sources for the

variability of the Ad26.ZEBOV, MVA-BN-Filo Ebola vaccine-induced immune responses between countries.

Assessing for other parasitic infections using more advances technologies, serology of viral infections such as CMV and Direct measurement of the microbiota present at the time of vaccination may provide additional information on the potential factors associated with variation in the response to vaccines in Africa [45].

## Supporting information

**S1 Appendix. Text A. Description of Sample Size calculation.** Table A. Description of the helminth ELISA commercial kits used, measured parasitic infection and cross reactivity. Table B. Overview of the substudy available sample selection by study (EBL2001, EBL2002, EBL3001), age group and country. Table C. Helminth ELISA test results at baseline among all participants of the helminth substudy (EBL2001, EBL2002, EBL3001 pooled). Table D. Characteristics of participants (pooled EBL2001, EBL2002, 3001) by helminth ELISA test result. Table E. Description of helminth ELISA test results in pooled EBL2001, EBL2002 and EBL3001 studies. Table F. Description of eosinophil levels among participants from studies EBL2001, EBL2002 and EBL3001. Table G. Description of eosinophil levels by any helminth ELISA test among participants from studies EBL2001 and EBL2002. Table H. Description of eosinophil levels in participants with any Helminth positive ELISA test from EBL2001 and EBL2002 studies (N = 55). Fig A. Description of the EBOV-GP antibodies at 21 days post dose 2 by eosinophil count. Fig B. Correlation between EBOV GP binding antibody geometric mean concentrations (GMC) and inflammatory markers. Fig C. Inflammatory markers (pg/mL) measured at baseline significantly associated with helminth test positivity (0 = No positive helminth ELISA test; 1 = any positive helminth ELISA test). Fig D. Inflammatory markers (pg/mL) statistically different between studies (EBL2001, EBL2002, EBL3001) at baseline. (DOCX)

## Acknowledgments

This Joint Undertaking receives support from the European Union's Horizon 2020 research and innovation programme and the EFPIA.

## Author Contributions

**Conceptualization:** Houreratou Barry, Cynthia Robinson, Viki Bockstal, Innocent Valea, Brian Greenwood, Rodolphe Thiébaut, Christine Lacabaratz.

**Data curation:** Houreratou Barry, Serge Somda.

**Formal analysis:** Houreratou Barry, Edouard Lhomme, Mathieu Surénaud, Serge Somda, Rodolphe Thiébaut, Christine Lacabaratz.

**Funding acquisition:** Cynthia Robinson, Nicolas Meda, Rodolphe Thiébaut.

**Investigation:** Houreratou Barry, Mathieu Surénaud, Moumini Nouctara, Christine Lacabaratz.

**Methodology:** Houreratou Barry, Edouard Lhomme, Innocent Valea, Rodolphe Thiébaut.

**Project administration:** Houreratou Barry.

**Resources:** Rodolphe Thiébaut.

**Software:** Serge Somda.

**Supervision:** Rodolphe Thiébaut, Christine Lacabaratz.

**Validation:** Cynthia Robinson, Rodolphe Thiébaut.

**Visualization:** Rodolphe Thiébaut.

**Writing – original draft:** Houreratou Barry, Edouard Lhomme.

**Writing – review & editing:** Houreratou Barry, Edouard Lhomme, Mathieu Surénaud, Moumini Nouctara, Cynthia Robinson, Innocent Valea, Serge Somda, Halidou Tinto, Nicolas Meda, Brian Greenwood, Rodolphe Thiébaut, Christine Lacabaratz.

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
