## [Decision Letter · Decision Letter 0]

6 Nov 2023

Dear Dr BARRY,

Thank you very much for submitting your manuscript "Helminth exposure and immune response to the two-dose heterologous Ad26.ZEBOV, MVA-BN-Filo Ebola vaccine regimen" for consideration at PLOS Neglected Tropical Diseases. As with all papers reviewed by the journal, your manuscript was reviewed by members of the editorial board and by several independent reviewers. The reviewers appreciated the attention to an important topic. Based on the reviews, we are likely to accept this manuscript for publication, providing that you modify the manuscript according to the review recommendations. 

Sincerely,

Eva Clark

Section Editor

Reviewer's Responses to Questions

**Key Review Criteria Required for Acceptance?**

**Methods**

-Are the objectives of the study clearly articulated with a clear testable hypothesis stated?

-Is the study design appropriate to address the stated objectives?

-Is the population clearly described and appropriate for the hypothesis being tested?

-Is the sample size sufficient to ensure adequate power to address the hypothesis being tested?

-Were correct statistical analysis used to support conclusions?

-Are there concerns about ethical or regulatory requirements being met?

Reviewer #1: - The authors have adeptly delineated both the primary and secondary objectives of the study.

- The study design demonstrates a high level of validity.

- The description of the study population is lucid and comprehensive.

- The calculated sample size aligns well with the scope of the study.

- The utilization of appropriate statistical analyses enhances the rigor of the study. However, it's important to note that while the authors state using RStudio for data analysis, it's worth clarifying that R programming was the actual tool employed. RStudio serves as a user-friendly interface for R programming and should be appropriately distinguished from the underlying analytical language. (Line 250, page9)

- On line 182, page 6, it was mentioned that participants from each initial study were randomly chosen. Could you please clarify if the number from each study was taken in proportion, or if all participants were combined and then selected randomly? This clarification would help readers better understand the process.

- Addressing a minor concern, it remains unclear if this study is an extension of the previously mentioned clinical trials, thus potentially obviating the need for new ethical approval. It would be beneficial for the authors to elaborate in more explicit terms on how they ensured adherence to ethical requirements in light of this potential continuity with previous trials.

**Results**

-Does the analysis presented match the analysis plan?

-Are the results clearly and completely presented?

-Are the figures (Tables, Images) of sufficient quality for clarity?

Reviewer #1: - The executed analyses are consistent with the authors' outlined methodology.

- The presentation of results is notably clear and comprehensible.

- Figure 1 is easy to comprehend, but its quality seems to be suboptimal. It might be beneficial to enlarge the numbers within the figure, as they appear quite small at present.

**Conclusions**

-Are the conclusions supported by the data presented?

-Are the limitations of analysis clearly described?

-Do the authors discuss how these data can be helpful to advance our understanding of the topic under study?

-Is public health relevance addressed?

Reviewer #1: - The conclusions drawn align logically with the presented data.

- The limitations of the study are distinctly articulated.

- Given that the study's outcomes indicate an absence of linkage between immune markers of helminth exposure and post-vaccination response, it would be advantageous for the authors to delve into potential factors contributing to the varying immunogenicity of the Ebola vaccine across diverse regions. This could serve to capture readers' attention and foster future research initiatives.

**Editorial and Data Presentation Modifications?**

Reviewer #1: I have no additional comments to contribute in this section.

**Summary and General Comments**

Reviewer #1: Barry and colleagues undertook a cross-sectional study aimed at assessing the correlation between immune markers indicative of helminth exposure and the immunogenic response following vaccination with Ebola vaccines. The study is well-written and focuses on a captivating topic. In brief, the research methods, results, and conclusions presented in the study are robust and well-founded. I have a few comments to contribute.

PLOS authors have the option to publish the peer review history of their article (what does this mean?). If published, this will include your full peer review and any attached files.

Reviewer #1: No

Figure Files:

Data Requirements:

Reproducibility:

References

---

## [Decision Letter · Decision Letter 1]

28 Feb 2024

Dear Dr BARRY,

We are pleased to inform you that your manuscript 'Helminth exposure and immune response to the two-dose heterologous Ad26.ZEBOV, MVA-BN-Filo Ebola vaccine regimen' has been provisionally accepted for publication in PLOS Neglected Tropical Diseases.

Best regards,

Eva Clark, M.D., Ph.D.

Section Editor

Eva Clark

Section Editor

Reviewer's Responses to Questions

**Key Review Criteria Required for Acceptance?**

**Methods**

-Are the objectives of the study clearly articulated with a clear testable hypothesis stated?

-Is the study design appropriate to address the stated objectives?

-Is the population clearly described and appropriate for the hypothesis being tested?

-Is the sample size sufficient to ensure adequate power to address the hypothesis being tested?

-Were correct statistical analysis used to support conclusions?

-Are there concerns about ethical or regulatory requirements being met?

Reviewer #1: The methodology section is well-crafted, addressing previous comments from peer reviewers effectively. The clarity and detail provided here lay a solid foundation for understanding the study's design and execution. I appreciate the effort to refine this section based on prior feedback, which has undoubtedly strengthened the manuscript. No further suggestions at this point.

**Results**

-Does the analysis presented match the analysis plan?

-Are the results clearly and completely presented?

-Are the figures (Tables, Images) of sufficient quality for clarity?

Reviewer #1: Improvements to the figures are notable and significantly enhance the reader's comprehension of the findings. The revised figures are now clear and effectively convey the study's results. This enhancement has substantially improved the manuscript's quality, allowing for a straightforward interpretation of the data presented. I commend your response to the feedback on earlier drafts, resulting in this improvement. No additional comments here.

**Conclusions**

-Are the conclusions supported by the data presented?

-Are the limitations of analysis clearly described?

-Do the authors discuss how these data can be helpful to advance our understanding of the topic under study?

-Is public health relevance addressed?

Reviewer #1: The conclusion is well-articulated, providing a compelling summary of the study's findings and their implications. The addition of more detailed information has not only made the manuscript more informative but also more engaging for readers. This section does an excellent job of capturing the study's significance and potential impact on the field. The authors' efforts to enhance this section based on previous reviews are evident and appreciated. No further recommendations for this section.

**Editorial and Data Presentation Modifications?**

Reviewer #1: I have no additional comment.

**Summary and General Comments**

Reviewer #1: The manuscript by Barry and colleagues, detailing a substudy nested within clinical trials, is commendably written and explores an intriguing topic: the association between participants’ immune responses to the Ebola vaccine and their history of helminthic parasites infection. I have no more comment to add.

In general, this manuscript is a significant contribution to the field, effectively bridging a gap in our understanding of the immune response to the Ebola vaccine in the context of helminthic parasite infections. The study is well-presented, and the revisions made in response to prior peer reviews have notably enhanced its clarity and depth. I look forward to seeing the final version published.

PLOS authors have the option to publish the peer review history of their article (what does this mean?). If published, this will include your full peer review and any attached files.

Reviewer #1: No

---

## [Editor Report · Acceptance letter]

3 Apr 2024

Dear Dr BARRY,

We are delighted to inform you that your manuscript, "Helminth exposure and immune response to the two-dose heterologous Ad26.ZEBOV, MVA-BN-Filo Ebola vaccine regimen," has been formally accepted for publication in PLOS Neglected Tropical Diseases.

Best regards,

Shaden Kamhawi

co-Editor-in-Chief

Paul Brindley

co-Editor-in-Chief
